# TRANSFORMERS SATISFY

## ABSTRACT

The Propositional Satisfiability Problem (SAT), and more generally, the Constraint Satisfaction Problem (CSP), are mathematical questions defined as finding an assignment to a set of variables such that all the constraints are satisfied. The modern approach is trending to solve CSP through neural symbolic methods. Most recent works are sequential model-based, and adopt neural embedding, i.e., reinforcement learning with graph neural networks, and graph recurrent neural networks. In this work, we propose Heterogeneous Graph Transformer (HGT), a one-shot model derived from the eminent Transformer architecture for factor graph structure to solve the CSP problem. We define the heterogeneous attention mechanism based on meta-paths for the self-attention between literals, the cross-attention based on the bipartite graph links between literals and clauses. Exploiting high-level parallelism, our model is able to achieve exceptional speed and accuracy on the factor graph for CSPs with arbitrary size. The experimental results have demonstrated the competitive performance and generality of HGT compared to the most recent baseline approaches.

## 1 INTRODUCTION

The Constraint Satisfaction Problems (CSP) is of central importance in several aspects of computer science, including theoretical computer science, complexity theory, algorithmics, cryptography, and artificial intelligence. CSP aims at finding a consistent assignment of values to variables such that all constraints, which are typically defined over a finite domain, are satisfied. In particular, there is an assortment of problems arising from artificial intelligence and circuit design that can be reduced to CSP subtypes, including map coloring, vertex cover, independent set, dominating set, and clique detection.

Solving a CSP on a finite domain is often an NP-complete problem with respect to the domain size. The conventional CSP-solvers rely on handcrafted heuristics that guide the search for satisfying assignments. These algorithms are focused on solving CSP via backtracking or local search. Hence, the resulted model is bounded by the greedy strategy, which is generally sub-optimal. With the advent of *Graph Neural Networks* (Scarselli et al. (2009)), the geometric deep learning (Bronstein et al. (2017)) for Non-Euclidean data has become one of the most emerging fields of machine learning. In particular, it brought deep learning solutions to one of the most dominant combinatorial optimization problems, the Constraint Satisfaction Problem (CSP) (Khalil et al. (2017)). Works including *NeuroSAT* (Selsam et al. (2018)) and *Circuit-SAT* (Amizadeh et al. (2018)) commenced the study of neural methods targeted at *CSP*. Later works, such as Yolcu & Póczos (2019) and You et al. (2019), attempted to solve CSP through different deep learning approaches. However, most pioneering works, such as neural approaches utilizing RNN or Reinforcement Learning, are still restricted to sequential algorithms, while clauses are parallelizable even though they are strongly correlated through shared variables.

In this work, we propose a hybrid model of the *Transformer* architecture (Vaswani et al. (2017)) and the Graph Neural Network for solving combinatorial problems, especially CSP. Our main contributions in this work are: (a) We derived meta-paths adopted from Sun et al. (2011) to formulate the message passing mechanism between homogeneous nodes (i.e., variable to variable, or clause to clause), which enable us to perform self-attention and let message pass through either variables sharing the same clauses, or clauses that include the same variables. We apply the cross-attention mechanism to optimize message exchanges between heterogeneous nodes(i.e., clause to variable, or variable to clause). (b) With the combination of *homogeneous attention* and *heterogeneous atten-*

*tion* mechanisms on bipartite graph structure, we then combine Transformer with Neuro-Symbolic methods to resolve combinatorial optimization on graphs. (c) We proposed *Heterogeneous Graph Transformer (HGT)*, a general framework for graphs with heterogeneous nodes. In this work, we trained the *HGT* framework to approximate the solutions of CSP (but not limited to CSP). Our model is able to achieve competitive accuracy, parallelism, and generality on CSP problems with arbitrary sizes

## 2 RELATED WORK

Recently, the Machine Learning community has seen an increasing interest in applications and optimizations related to constraint satisfaction problem solving. Various frameworks utilising diverse methodologies have been proposed, offering new insights into developing CSP solvers and classifiers. For example, Bello et al. (2016) adopts Reinforcement Learning in their *Neural Combinatorial Optimization*, with an approach based on policy gradients. On the other hand, works such as Evans et al. (2018) and Arabshahi et al. (2018) have demonstrated the effectiveness of recursive neural networks in modeling symbolic expressions. Meanwhile, Prates et al. (2019) proposed an embedding-based message-passing algorithm for solving Traveling Salesman Problem (TSP), a highly relevant CSP problem.

*NeuroSAT* (Selsam et al. (2018)) is a graph neural network model that aims at solving the Boolean Satisfiability Problem (SAT) without leveraging the greedy search paradigm. It approaches SAT as a binary classification problem during training and finds an SAT assignment from the latent representations during inference. *NeuroSAT* is able to search for solutions to problems of various difficulties despite training for relatively small number of iterations. As an extension to this line of work, Selsam & Bjørner (2019) proposes a neural network that facilitates variable branching decision making within high-performance SAT solvers on real problems.

*PDP* Amizadeh et al. (2019) is a generic neural framework for learning CSP solvers based on the idea of Propagation, Decimation, and Prediction. *PDP* provides a completely unsupervised training mechanism for solving SAT via energy minimization, and can be seen as learning optimal message passing strategy on probabilistic graphical models.

*G2SAT* (You et al. (2019)) is a deep generative framework that learns to generate SAT formulas from a given set of input formulas while preserving the graph statistics. Even though *G2SAT* lacks the ability to derive solutions, it provides synthetic formulas for hyperparameter optimization.

*RLSAT* (Yolcu & Póczos (2019)) learns SAT solvers through deep reinforcement learning and iterative refinement. It incorporates a graph neural network into a *Stochastic Local Search(SLS)* algorithm to act as the variable selection heuristic during training. However, since *SLS* begins with randomly initialized parameters, and a non-zero terminal reward is given only when a satisfying assignment is found, *RLSAT* requires a curriculum learning process for performance improvement. As a result, *RLSAT* becomes inefficient when its learning process starts with large complex graphs, in which satisfying assignments are hard to obtain. In our *HGT*, each possible state of assignment corresponds to a likelihood, which can be minimized to train the model. In particular, *HGT* is able to achieve optimal performance in efficiency and accuracy regardless of input graph sizes.

## 3 BACKGROUND

### 3.1 CONSTRAINT SATISFACTION PROBLEMS

Constraint Satisfaction Problems (CSP) (Kumar (1992)) is a fundamental problem in logic study that constitutes the cornerstone of combinatorial optimization. It provides feasible models to real world applications and is intensively involved in the design of artificial intelligence. An instance of CSP problem, $CSP(V, U)$, is constituted of two main components: a set of $N$ variables $V = \{v_i \in D : i \in 1...N\}$, defined over a discrete domain $D$; and a set of $M$ constraint functions or factors, $U = \{u_j(q_j) : j \in 1...M\}$, where $q_j$ is a subset of $V$ subject to $u_j$. For each $u_j \in U$, $u_j : D^{|q_j|} \rightarrow \{0, 1\}$ outputs 1 if the input $q_j$ satisfies constraint $u_j$, and 0 otherwise. A CSP problem can be formulated in *Conjunctive Normal Form (CNF)* (Pfahringer (2010)) with the goal of finding an assignment of variables that satisfies all constraints. For a given assignment to $V$, the measure of

a CSP problem, $\phi : \{0, 1\}^N \rightarrow \{0, 1\}$, is a function that returns 1 if all the clauses in the problem are satisfied, and 0 otherwise. Such a measure function is in *CNF* if

1. For all $u_j \in U$, $u_j$ is a disjunction of literals. Here literals refer to $v_i$ or $\neg v_i$, for $v_i \in q_j$;
2. $\phi(V) = \wedge u_j, j = \{1, ..., M\}$.

## 3.2 GRAPH NEURAL NETWORKS AND GRAPH ATTENTION NETWORKS

*Graph Neural Network* (GNN) (Scarselli et al. (2009)) provides a basic architecture to operate neural network on graphs. Suppose $V$ is a set of nodes, where each vertex $v_i \in V$ carries a node feature $h_i \in \mathbb{R}^{d_V}$, and $\mathcal{E} \subset V \times V$ is a set of undirected edges, where each edge $e_{i,j} \in \mathcal{E}$ that connects $v_i$ to $v_j \in V$ carries an edge feature $h_{i,j} \in \mathbb{R}^{d_{\mathcal{E}}}$. For time step $t \in \{1, ..., T\}$, a GNN on the graph $G(V, \mathcal{E})$ passes messages between direct neighbors, and iteratively updates the feature vector of each $v_i \in V$ following:

$$h_{v_i}^t = Q \left( h_{v_i}^{t-1}, \bigcup_{v_j \in \mathcal{N}_i} M^t(h_{v_i}^{t-1}, h_{v_j}^{t-1}, h_{i,j}) \right) \tag{1}$$

where $\mathcal{N}_i = \{v_j \in V : \exists e_{i,j} \in \mathcal{E}\}$, and $M$, $Q$ are the message function and the update function to be learned, respectively.

## 3.3 TRANSFORMER AND THE RELATION WITH GNN

Transformer (Vaswani et al. (2017)) is the state-of-the-art approach for sequential data and transduction problems that relies on self-attention mechanism. When looking into the architecture of *Transformer*, the computation of similarity between queries and keys highly resembles finding a correlation between two types of nodes in fully connected bipartite graphs, while the self-attention mechanism among queries or keys (or tokens in *NLP* terminology) can be seen as calculating the weighted contribution in a half-complete connected graph, where an output token does not depend on future words. With these resemblance, we want to extend the *Transformer* architecture to heterogeneous graph structures, such as bipartite graphs.

## 4 METHODOLOGY

### 4.1 FACTOR GRAPHS AND META-PATHS

The value of each constraint is binary, which makes it possible to express the measure of a *CNF* as:

$$\phi(V, U) = \prod_{j=1}^{M} u_j(q_j) \tag{2}$$

The expression can be properly presented as an undirected bipartite factor graph. We construct such a factor graph $G((V, U), \mathcal{E})$ by defining the set of variables $V = \{v_1, \ldots, v_n\}$, the set of clauses $U = \{u_1, \ldots, u_m\}$, and edges $\mathcal{E}$ by: $e_{i,j} \in \mathcal{E}$ iff variable $v_i$ is involved in constraint $u_j$ either in positive or negative relation. In Yolcu & Póczos (2019), each edge is assigned with a type depending on the polarity of the variable it connects to. The positive occurrence of a variable $v_i$ in a clause (or factor) $u_j$ is represented with the positive sign $(+)$, whereas its negative occurrence $\neg v_i$ in $u_j$ is represented with the negative sign $(-)$. Hence, a pair of $n \times m$ bi-adjacency matrix $\mathbf{A} = (A_+, A_-)$ is used to store two types of edges such that $A_+(i, j) = 1 \Leftrightarrow v_i \in u_j$ and $A_-(i, j) = 1 \Leftrightarrow \neg v_i \in u_j$. Here $v_i \in u_j$ implies that $v_i$ instead of its negation $\neg v_i$ is directly involved in $u_j$. Each edge $e_{i,j} \in \mathcal{E}$ is then assigned a value equals to 1 for edges in $A_+$ and $-1$ for edges in $A_-$. With the factor graph representation, graph neural network can be applied as a CSP-solver (Selsam et al. (2018); Yolcu & Póczos (2019)). However, due to the fact that factor graph is bipartite in which variables are only connected to clauses, clauses are only connected to variables. Furthermore, traditional *GNN* only passes messages to local neighbors, and is thus inefficient at passing messages between variables or between clauses. Consequently, variable nodes update their state solely based on their current states and the states of the clauses they affect, vice versa. In this work, we propose to pass message through length-2 meta-paths in addition to existing edges, which enables: 1) variables

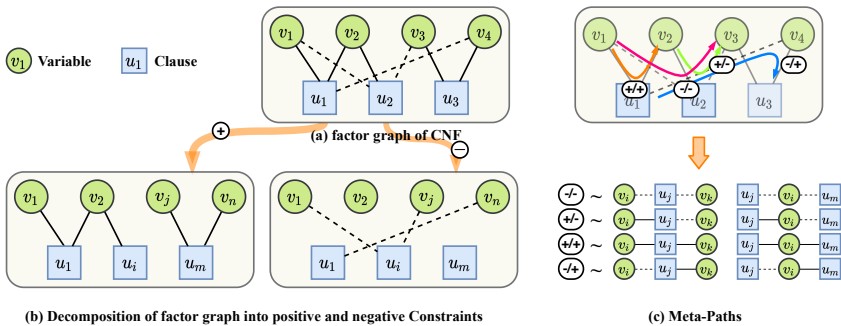

Figure 1: (a) Factor graph for the *CNF* with measure $\phi = (v_1 \vee v_2 \vee v_4) \wedge (\neg v_1 \vee v_2 \vee \neg v_3) \wedge (v_3 \vee v_4)$, where solid lines are the positive incidences of $v_i$ in $u_j$, and dashed lines are the negative incidences of $\neg v_i$ in $u_j$; (b) the decomposition of the factor graph according to the positive and negative relations, which are used for crossing-attention; (c) meta-paths are used for the self-attention among literals or clauses.

to incorporate the information from variables that share the same clauses during state updates; 2) clauses to incorporate the information from clauses that share some variables when update their states. In a *CSP* factor graph, we define that a meta-path $m_{i,j} = (v_i, v_k, v_j)$ between nodes $v_i$ and $v_j$ exists if there exists some $v_k \in V$ s.t. $\exists e_{i,k} \in \mathcal{E}$ and $\exists e_{k,j} \in \mathcal{E}$. Since self-attention mechanism is not symmetric, our meta-path is directed. As a result, we get four types of meta-paths in total (i.e., $\{(+, +), (+, -), (-, +), (-, -)\}$), as illustrated in Figure 1 (c). The adjacent matrix of such a meta-path can be easily computed by matrix multiplication of $A_+$ and $A_-$ or their transposes. Take $A_{(+,+)}$, $A_{(+,-)}$ as examples, the adjacency matrix $A_{(+,+)} = A_+ A_+^T$ stores all $(+, +)$ meta-paths, and $A_{(+,-)} = A_+ A_-^T$ stores all $(+, -)$ meta-paths. A diagonal entry $A_{(+,+)}[i, i]$ indicates the number of positive edges that $v_i$ has, and an off-diagonal entry $A_{(+,+)}[i, j]$ indicates the existence of $(+, +)$ meta-path from $v_i$ to $v_j$.

## 4.2 HOMOGENEOUS AND HETEROGENEOUS GRAPH ATTENTION CONVOLUTION

In order to incorporate attention mechanism into the bipartite graph, we extend the original Graph Attention Network (GATConv) (Veličković et al. (2017)) with *heterogeneous graph attention*, while we consider the original attention in GATConv as *homogeneous graph attention*. Heterogeneous attention is used for nodes of different kinds. Nodes in different partitions are normally of different features or characteristics, e.g., literals forming a partition different from clauses in their partition. Suppose literal and clause feature matrices are $V \in \mathbb{R}^{N \times F_v}$ and $U \in \mathbb{R}^{M \times F_u}$, respectively; and their adjacency matrix $A \in \mathbb{R}^{M \times N}$. The **heterogeneous attention** is define as:

$$e_{v_i,u_j} = \sigma([a_v || a_u]^T [W_v \overrightarrow{v_i} || W_u \overrightarrow{u_j}]), \quad e_{u_j,v_i} = \sigma([a_u || a_v]^T [W_u \overrightarrow{u_j} || W_v \overrightarrow{v_i}])$$

$$\alpha_{v_i,u_j} = \text{softmax}_{u_j}(e_{v_i,u_j}) = \frac{exp(e_{v_i,u_j})}{\sum_{u_k \in N(v_i)} exp(e_{v_i,u_k})}$$

$$\alpha_{u_j,v_i} = \text{softmax}_{v_i}(e_{u_j,v_i}) = \frac{exp(e_{u_j,v_i})}{\sum_{v_k \in N(u_j)} exp(e_{u_i,v_k})} \quad (3)$$

$$V' = ReLU(VW_v + \mathcal{A}_{u \to v} UW_u), \quad U' = ReLU(UW_u + \mathcal{A}_{v \to u} VW_v)$$

where $W_v \in \mathbb{R}^{F_v \times F}$ and $W_u \in \mathbb{R}^{F_u \times F}$ are embedding weights for variables and clauses that transform input feature into a common high-level feature of dimension $F$. $a_v \in \mathbb{R}^F$ and $a_u \in \mathbb{R}^F$ are shared attention mechanism on variables and clauses. Attention matrix $\mathcal{A}_{u \to v} \in \mathbb{R}^{N \times M}$ stores edge scores from clauses to variables as $\mathcal{A}_{u \to v}[i, j] = \alpha_{u_j,v_i}$; and $\mathcal{A}_{v \to u} \in \mathbb{R}^{M \times N}$ stores edge scores from variables to clauses as $\mathcal{A}_{v \to u}[j, i] = \alpha_{v_i,u_j}$. $\sigma$ is non-linearity function, i.e., LeakyReLU used in GATConv (Veličković et al. (2017)).

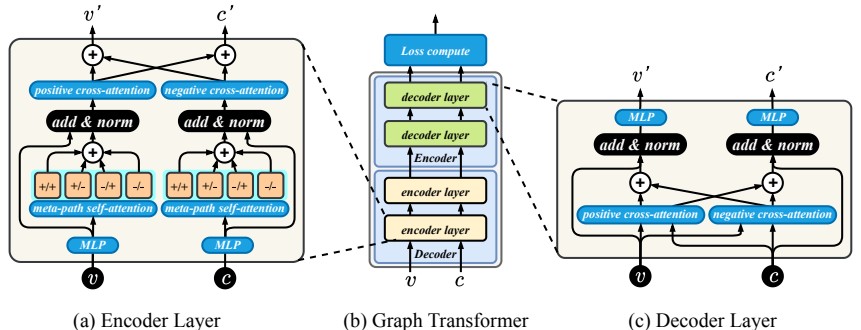

|  |  |  |
|---|---|---|
| (a) Encoder Layer | (b) Graph Transformer | (c) Decoder Layer |

Figure 2: Our *Heterogeneous Graph Transformer* architecture consists of a set of encoders and decoders connected sequentially, as in (b). Its encoder and decoder architectures are shown in (a) and (c), respectively.

### 4.3 HETEROGENEOUS GRAPH TRANSFORMER (HGT)

Our model *HGT* adopts an encoder-decoder structure similar to Britz et al. (2017), with four encoder-layers in the encoder, and three decoder-layers in the decoder. It is flexible to adjust the number of encoder and decoder layers.

**Encoder**. Prior to entering the stack of encoder-layers, the input node features are embedded into high-dimensional space with *MLP*. Within each encoder-layer, every graph node first aggregates the message (or information) from nodes of its kind through *meta-paths*. Note that a node (variable or clause) of bipartite graph has no direct connection within homogeneous nodes. Messages can only get passed among homogeneous nodes through *meta-paths*. We emphasize such type of communication between nodes of the same kind as *self-attention*, which is implemented with *homogeneous attention* mechanism. The weighted messages are passed between variables and clauses through the *cross-attention* mechanism, implemented with the *heterogeneous attention mechanism*. The two types of attention are connected through the residual block and *layer normalization* (Ba et al. (2016)), as shown in Figure 2 (a).

**Decoder**. Decoder consists of several decoder-layers. Inside each decoder-layer, there is one cross-attention module followed by residual connection and *layer normalization*, as in Figure 2 (c). The attention-weighted node features are then fed into the *MLP* for dimension reduction.

#### 4.3.1 LOSS EVALUATION

For a given CSP$(V, U)$, each combination of variable assignments corresponds to a probability. The original measure $\phi(V, U)$ is a non-differentiable staircase function defined on a discrete domain. $\phi(V, U)$ evaluates to 0 if any $u_j \in U$ is unsatisfied, which disguises all other information including the number of satisfied clauses. For training purpose, a differentiable approximate function is desirable. Therefore, the proposed model generates a continuous scalar output $x_i \in [0, 1]$ for each variable, and the assignment of each $v_i$ can be acquired through:

$$v_i = \left\lfloor \frac{x_i}{0.5 + \epsilon} \right\rfloor \tag{4}$$

where $\epsilon$ is a small value to keep the generated $v_i$ in $\{0, 1\}$. With continuous $x_i, i = 1, ..., N$, we can approximate disjunction with $max(\cdot)$ function and define $\phi(\cdot)$ as

$$\phi(x_1, ..., x_N) = \prod_{j=1}^{M} \max(\{l(x_i) : v_i \in q_j\}) \tag{5}$$

Here, the literal function $l(x_i, e_{ia}) = \frac{1 - e_{ia}}{2} + e_{ia} x_i$ is applied to specify the polarity of each variable. Inspired by Amizadeh et al. (2019), we replace the *max* function with a differentiable *smoothmax*$(\cdot)$

$$S_\tau(x_1, ..., x_N) = \frac{\sum_{i=1}^{n} x_i e^{\tau x_i}}{\sum_{i=1}^{n} e^{\tau x_i}} \tag{6}$$

Mathematically, $S_\tau(x_1, ..., x_N)$ converges to $\max(x_1, ..., x_N)$ as $\tau \to \infty$. However, since the assignment of each $x_i$ is continuous and the threshold is $0.5$, *HGT* doesn't need a strict *max* function to judge if a clause contains a literal with value $1$. We note that $\tau = 5$ is enough for our model in practice. By maximizing the modified $\phi$, *HGT* is trained to find the optimal assignment for each CSP problem. For numerical stability and computational efficiency, we train *HGT* by minimizing the *negative log-loss*

$$\mathcal{L}(x_i, ..., x_N) = -log(\phi) = -\sum_{j=1}^{M} log(S_\tau(\{l(x_i) : v_i \in q_j\})) \tag{7}$$

During the training process, we realize that the value for each $x_i$ is trending towards $0.5$ to minimize the loss function. We then append a regularizer

$$r(u_i, ..., u_M) = -\sum_{j=1}^{M} ReLU(u_j - 0.5) \tag{8}$$

to punish those clauses with values near $0.5$, while allowing clauses with small values to increase. In this way, the regularizer encourages the value of each $x_i$ to approach $0$ or $1$, without disturbing the maximization of the values of clauses that $x_i$ is connected to.

## 5 EXPERIMENTAL EVALUATION

### 5.1 DATASET

In order to learn a CSP solver that can be applied to various classes of satisfiability problems, we drew our training set from four classes of problems with distinct distributions: random 3-SAT, graph coloring, vertex cover, and clique detection.

| Class | Distribution | Variables (n) | Clauses (m) / Edges (p) |
|---|---|---|---|
| **Random 3-SAT** | $rand_3(n, m)$ | $n = \{100, 150, 200\}$ | $m = \{430, 645, 860\}$ |
| $k$-**coloring** | $color_k(N, p)$ | $n = k \times N$ for $N=\{5, 10\}$ | $p = 50\%$ |
| $k$-**cover** | $cover_k(N, p)$ | $n=(k + 1) \times N$ for $N=\{5, 7\}$ | $p = 50\%$ |
| $k$-**clique** | $clique_k(N, p)$ | $n = k \times N$ for $N=\{5, 10\}$ | $p = \{20\%, 10\%\}$ |

Table 1: The summary of our chosen dataset. For random k-SAT problems, *n* and *m* refer to the number of variables and clauses. For graph problems, *N* is the number of vertices, *k* is the problem-specific parameter, and *p* is the probability that an edge exists.

For the random 3-SAT problems, we drew a total of 1200 synthetic SAT formulas from the *SATLIB* benchmark library (Hoos & Stützle (2000)). These graphs consist of variables and clauses of various sizes, and should reflect a wide range of difficulties. For the latter three graph problems, we sampled 4000 instances from each of the distributions that are generated according to the scheme proposed in Yolcu & Póczos (2019). In particular, we tuned the problem-specific parameters for each set of instances to assess our model's ability to generalize to other CSP problems. All formulas in our dataset, regardless of distribution, are satisfiable, and are encoded in DIMACS Conjunctive Normal Form (CNF), the natural format for SAT problems. Models trained on these graphs generalizes to other CSP problems, datasets, and larger graphs.

### 5.2 BASELINES

**Baseline models**. To assess the validity of our model, we compared our framework against three categories of baselines: (a) the classic stochastic local search algorithm for CSP solving - **WalkSAT** (Selman et al.), (b) the reinforcement learning-based CSP solver - **RLSAT** (Yolcu & Póczos (2019)), (c) the generic but innovative neural framework for learning CSP solvers - **PDP** (Amizadeh et al. (2019)). Among these baselines, **PDP** falls into the same category as the proposed model, which utilizes the one-shot algorithm.

**Baseline setting**. For the RLSAT and PDP baseline models, we strictly preserved their default parameters in order to reproduce the original performances. Specifically, the RLSAT model was

|  | $color_3$ | $color_3$ | $cover_2$ | $cover_3$ | $clique_3$ | $clique_3$ |
|---|---|---|---|---|---|---|
|  | $(5, 0.5)$ | $(10, 0.5)$ | $(5, 0.5)$ | $(7, 0.5)$ | $(5, 0.2)$ | $(10, 0.1)$ |
| RLSAT | 99.01% | 71.49% | 93.78% | 92.04% | 95.72% | 97.96% |
|  | $\pm 9.93\%$ | $\pm 20.92\%$ | $\pm 12.12\%$ | $\pm 13.76\%$ | $\pm 15.40\%$ | $\pm 12.35\%$ |
| Our Model | **87.51%±1.45%** | | **97.77%±0.11%** | | **97.35%±0.37%** | |

Table 2: Our model's performance during training, compared to that of the baseline model RLSAT. We trained both models with 500 epochs on six distributions of graph problem sets. Within each cell, we present the metric of accuracy in the format of: [average accuracy %]±[standard deviation %].

trained on their self-generated datasets via curriculum learning. For the PDP framework, we tested a fully neural CSP solver with $weight\_decay = 10^{-10}$ and $dropout\_rate = 0.2$.

## 5.3 EXPERIMENTAL SETTING

**Software**. The PyTorch implementation of our model is open-sourced, and the code is available at https://git.io/JUybf. Our model employs PyTorch Geometric (Fey & Lenssen (2019)), and is able to achieve high efficiency in both training and testing by taking full advantage of GPU computation resources via parallelism.

**General setup**. Our model for the experiments discussed in this section is configured as follows. Structures are implemented according to the architecture presented in Figure 2. For the encoder, we adopted four *encoder-layers* with the number of channels set to {16, 32, 32, 32}; for the decoder, we adopted three *decoder-layers* with the number of channels set to {32, 16, 16}. Optimizer Adam(Kingma & Ba (2017)) with $\beta_1 = 0.9$, $\beta_2 = 0.98$, $\epsilon = 10^{-9}$ was applied to train the model. Our learning rate varies with each step taken, and follows a pattern that is similar to the one adopted by Noam (Vaswani et al. (2017)).

$$learning\_rate = \beta * \min(step\_num^{-0.5}, step\_num \times warmup\_steps^{-1.5}) \qquad (9)$$

Specifically, our learning rate increases linearly for the initial warm-up steps, then gradually decay at a rate that is inversely proportional to the square root of the step number. $\beta$ is the coefficient used to adjust the scale of learning rate. To ensure fairness of comparison, we adopted a dropout rate of 0.2 for regularization, which is the same value as Amizadeh et al. (2019).

## 5.4 RESULTS AND EVALUATION

**Validity of training**. Table 2 summarizes the performance of our model, as compared to that of RL-SAT. For the training process of each model, we strictly followed the setting as explained in Section 5.2 and 5.3. Since our model adopts a semi-supervised training strategy, and is capable of processing graphs of arbitrary size, we were able to combine numerous distributions of the same problem class into one single dataset during training, regardless of the problem-specific parameters. We observed that our model's average accuracy is higher than that of RLSAT in all six distributions. For further analysis, we present the holistic learning curves in Figure 3, where the shaded areas visualize the standard deviations of each model's validation scores. From the figure, we noticed that for the latter 100 epochs, *RL-KC* and *RL-KV*'s validation performance start to oscillate significantly. Investigating the characteristics of *Reinforcement Learning*, we discovered that RLSAT, upon encountering graphs with new scales, performs a whole new process of exploration. Therefore, RLSAT fails to generalize its learnt experience to subsequent larger graphs, which results in an unstable validation score during training. In comparison, our model adopts a highly paralleled message-passing mechanism, which updates all nodes of all graphs simultaneously at each epoch. As a result, we are able to reach a high score rapidly, while maintaining the accuracy with little fluctuation.

We understand that the graph problems generated by RLSAT may not be sufficient in size and difficulty to fully assess the capability of our model. Hence, in addition to testing on a diversified distribution of graph problems, we also experimented the validity of our model on the classic random 3-SAT dataset, and compared our results with that of PDP. As seen in Table 3, our model retains the ability to achieve high accuracy with little time required for processing each graph. In comparison, PDP takes a significantly longer time for validation, while reaching an average accuracy that fails to exceed ours.

| | $rand_3(100, 430)$ | | $rand_3(150, 645)$ | | $rand_3(200, 860)$ | |
|---|---|---|---|---|---|---|
| | Time (s) | Acc (%) | Time (s) | Acc (%) | Time (s) | Acc (%) |
| PDP | 0.0743 | 96.51±0.69 | 0.0413 | 95.50±0.23 | 0.0915 | 93.65±0.62 |
| Our Model | **0.00368** | **97.06±0.28** | **0.00361** | **96.80±1.31** | **0.0128** | **96.19±1.57** |

Table 3: Our model's performance during training, compared to that of the baseline model PDP. We train both models with 1000 epochs on three distributions of random 3-SAT problems. We present the validation time of each graph, as well as the validation accuracy, which is in the format of: [average accuracy%]±[standard deviation%].

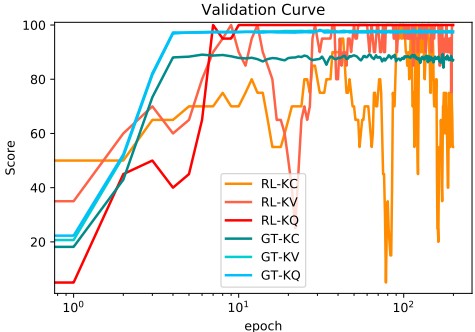
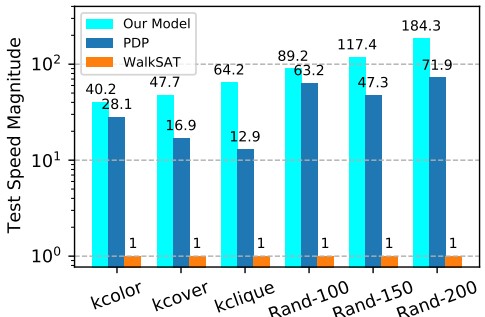

Figure 3: The learning curve of HGT (GT) and that of RLSAT (RL). The $x$-axis measures the number of epochs trained. The y-axis measures the validation score in percentage. Both models are trained on {KC: $color_3(10, 0.5)$, KV: $cover_3(7, 0.5)$, KQ: $clique_3(10, 0.1)$}.

Figure 4: Our model's test speeds compared to those of PDP, with the speedup of WalkSAT as the baseline metric. The x-axis indicates the six data distribution, upon which we test the models. The y-axis measures the speedup of the models, w.r.t. WalkSAT's performance.

**Efficiency of testing**. Noting the discrepancy between the validation time, we further analyzed the efficiency of our model by measuring the time taken for each trained model to test on an individual graph. To diversify our test sets, we chose 2000 instances from each of the following distributions: {$color_3(20, 0.5)$, $cover_3(9, 0.5)$, $clique_3(20, 0.05)$ }, as well as 100 instances from each of the random 3-SAT distributions: {$rand_3(100, 430)$, $rand_3(150, 645)$, $rand_3(200, 860)$}. Summarized in Figure 4 are the results we obtained from comparing the average test speeds of our model against that of PDP, with the performance of WalkSAT as the metric. As demonstrated in the figure, our model is capable of achieving higher average test speeds regardless of the graph structure. This observation can be explained by the fact that our model allows communication within homogeneous nodes, which provide all nodes with abundant semantic information when updating their states. Therefore, our model requires fewer iterations of message passing, and achieves greater efficiency.

## 6 CONCLUSION

In this paper, we proposed *Heterogeneous Graph Transformer (HGT)*, a one-shot model derived from the eminent Transformer architecture for factor graph structures, to solve the CSP problem. We defined the homogeneous attention mechanism based on meta-paths for the self-attention between literals or clauses, as well as the heterogeneous cross-attention based on the bipartite graph links from literals to clauses, or vice versa. Our model achieved exceptional parallelism and accuracy on the factor graph form of CSPs with arbitrary sizes. The experimental results have demonstrated the competitive performance and generality of *HGT* in several aspects. Future efforts in this direction would include the extraction of high-level embeddings from *HGT* as well as the application of our proposed algorithm to further aid classic CSP solvers on solving combinatorial optimization problems. Our work also suggests the possibility of transforming classic techniques based on *Conflict-Driven Clause Learning* solvers to neural frameworks.

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
