# OpenReview forum: "Transformers satisfy"
_ICLR.cc/2021/Conference — Reject_

### Official Review · AnonReviewer2 · 2020-10-19
**Improvement of GNNs on synthetic SAT problems**

**Rating:** 4
**Confidence:** 4

**Review:**

Summary: The paper "Transformers satisfy" presents an improved graph neural network model to solve SAT problems. Prior applications of GNNs to SAT have used convolutional GNNs instead of Graph Attention Networks (GATs), and this work suggests a modification of GATs to improve their performance on the bipartite graphs encountered in SAT.

The main part of the modification is that their GNNs consider up to two hops between nodes, such that variables can attend to other variables, and clauses can attend to other clauses. The paper then also suggests a way to allow "cross-attention" between clauses and variables.

The authors report moderate accuracy gains at predicting satisfiability on small synthetic SAT formulas.

On the positive side, the authors explore the use of "meta-paths", a way to exploit the bipartite structure to speed up the GNNs. But the authors do not contrast that with plain GATs applied to their problem so it is hard to say how much that idea contributes.


My concerns are as follows:
- The paper claims to address constraint satisfaction problems (CSP), but focuses on SAT.
- The paper claims to consider Transformers, but uses graph neural networks.
- Evaluation done on very synthetic benchmarks. Does not necessarily carry over to other benchmarks.
- Table 2 shows that the performance of their approach does not outperform their baseline in 2 out of 3 benchmarks. Table 3 shows moderate gains, but only on random SAT problems.
- The motivation of the work is to improve the performance of neural networks on SAT problems, but I am not convinced that there is any perspective to beat CDCL solvers with neural networks at this problem.


Writing style: The paper is understandable, but could be written better. Formulations in several places are unclear. For example:
- Abstract: "We define the heterogeneous attention mechanism based on meta-paths for the self-attention between literals, the cross-attention based on the bipartite graph links between literals and clauses." What does that mean? What are meta paths? What is cross-attention?
- Introduction, first sentence: What does it help the reader to see only the claim that CSP is important. If you feel that SAT and CSP need motivation, please provide concrete applications.
- Introduction: "We apply the cross-attention mechanism to optimize message exchanges between heterogeneous nodes" Most of the terms here were not yet introduced and I only understood this sentence after reading the rest of the paper.
- The main contribution (Section 4.2) is not explained in detail, only the math is given and the reader has to reconstruct the motivation.


Minor comment:
- Check capitalization of references.

---

### Official Review · AnonReviewer3 · 2020-10-28
**A promising idea which requires stronger empirical analysis**

**Rating:** 4
**Confidence:** 3

**Review:**


### Summary

The paper presents a model for inferring the solution of a constraint satisfaction problem over boolean variables expressed in *Conjunctive Normal Form (CNF)*. The proposed model builds on existing works which represent the factor graph of the CNF as a bipartite graph, and use a graph neural network for the learning task. It re-defines the graph to facilitate the message passing among the nodes which are in the same part of the graph (i.e. CNF variables/clauses). It also introduces a cross-attention mechanism between the nodes belonging to different parts.

The proposed model is evaluated on a set of benchmark SAT instances and is compared with an existing baseline method.

### Strong points

- The preliminary results are encouraging
- The ideas seem to be novel, and there is a potential for impact
- The paper is well-written and easy to follow

### Aspects to be improved

**Ablation study**: It seems that meta-paths can be applied without the attention mechanism. If this is the case, it should be clarified that to what extent the obtained results depend on the combination of both ideas.

**Instance selection**: SAT instances vary significantly in terms of difficulty. It seems that the problem instances used in the experimental evaluation are all *easy* instances. I was not able to evaluate the difficulty of the instances from the dataset description. To have an estimate of the difficulty of the problems, I solved all instances in the *Random 3-SAT* class using the CDCL solver *MiniSAT* and obtained these runtimes:

$$
\begin{array}{lrrr}
\textbf{problem class} & \textbf{\#instances} & \textbf{total runtime} & \textbf{average runtime} \\\\
rand_3(100, 430) & 1000           & 0.709      & 0.00071   \\\\
rand_3(150, 645) & 100            & 0.570      & 0.00570   \\\\
rand_3(200, 860) & 100            & 6.171      & 0.06171
\end{array}
$$

These results indicate that the *Random 3-SAT* instances (especially the ones with 100 and 150 variables) are particularly easy SAT instances. The results of the experiments in the paper leads me to believe that the instances of other problem classes (i.e. $k$-coloring, $k$-cover, and $k$-clique) are also among the easier SAT instances. The reported accuracies in Tables 2 and 3 are quite high (both for the proposed method and the baseline). This indicates that there is room for including more challenging instances in the empirical analysis.

Note that I am not suggesting that the learned models should compete with standard SAT solvers. I'm simply using standard solvers as a proxy to estimate the difficulty of the instances. The authors can use a similar approach to categorize a set of instances (e.g. a subset selected from a recent SAT competition) into different levels of difficulty (e.g. easy, medium, and hard). This will make the empirical analysis more reliable (and ideally further demonstrates the advantages of the proposed method).

**Analysis and reporting**:

The space in the *experimental evaluation* section can be used more effectively. In contrast to the detailed description of the instances, baseline, and setup, the main results are aggregated into two small tables. I suggest spending this space on a more extensive empirical analysis. Moreover, the section and figure on *efficiency of testing* can be summarized in a few sentences. If the authors insist on presenting the inference speed as a merit of their proposed approach, they should identify instances where the solving time using a SAT solver is prohibitively long (certainly more than a fraction of a second).

### Recommendation

The paper presents interesting ideas, which can potentially extend to domains beyond SAT solving. However, the proposed ideas need to be evaluated in a more extensive and rigorous empirical study.

### Additional suggestions for improvement

- An interesting question (which is for example studied by *RLSAT*) is the performance of models trained on one problem class when evaluated on another class. This is not a central question, yet would be a nice experiment to be included in an appendix.

---

### Official Review · AnonReviewer1 · 2020-10-29
**Some good ideas but lots of needed improvements**

**Rating:** 3
**Confidence:** 5

**Review:**

Summary of the paper:
The paper proposes a novel architecture called Heterogeneous Graph Transformers (HGT) to solve Constraint Satisfaction Problems (CSP) using unsupervised learning. Their model seems to achieve good results on classical problems encoded as CSP (3-SAT, k-coloring, k-cover, k-clique) with a few hundreds of variables and clauses, along with faster running time as compared to other contemporary methods because of a clause-parallel model architecture. The main contribution of the paper is to modify typical message-passing steps in graph neural networks to take more of the clause-variable structure into account, and accelerate message-passing in-between clauses and variables, respectively.

Strengths:
- A new approach to solving CSP using Graph Neural Networks/Transformers with some modeling improvements leading to reduced solution times and better success rate in finding satisfiable assignments.

Weaknesses:
- The experimental setup and results need significant improvements before this work can be published; see details in Questions below.

- Writing: The motivating story in the introduction is not so clear-cut. There is a mention of previous work being limited to sequential algorithms, but no strong intuition is offered for why clause-parallelism is useful. Additionally, some key technical terms (e.g., "cross-attention", "meta-paths") are mentioned with little context, making it difficult for the reader to start grasping the paper's contributions early on.

- Novelty: The attention mechanisms presented are very much similar to other attention mechanisms like Selsam et al. (2018). The use of transformer-type models is not uncommon, and was used prominently for TSP-type problems in this paper (which was not cited): Kool, Wouter, Herke van Hoof, and Max Welling. "Attention, Learn to Solve Routing Problems!." International Conference on Learning Representations. 2018.


Recommendation:
There are some positives to this work, particularly in the careful message-passing design. However, I have to recommend a Reject because the experimental results are very, very limited at this point. It is unlikely that the authors will be able to address all the issue I raise here, but I encourage them to do so in the near future and submit to future conferences.


Questions to the authors (in no particular order):
1- Please show comparisons with a state-of-the-art optimization CSP solver such as IBM CPLEX's CP Optimizer or others.

2- Test accuracy: Figure 4 only showcases the speedup achieved by the method compared to other approaches. Please report the test accuracy numbers (mean and standard deviation). Otherwise, the tables show only training results which does not tell the reader much.

3- Homogeneous and heterogeneous graph attention: The distinctions between homogeneous and heterogenous graph attention is not clearly stated. The only distinction seems to be different sizes of the initial node (v in V) and edge (u in U) embedding sizes; v is of F_v dimension and u is of F_u dimension as opposed starting with same dimensions as in Velickovic et al 2017. Ideally, in heterogeneous attention, e_vi_uj and e_uj_vi should lead to different values. However, based on the definition in Equation (3), both of them will attain the same value. It would be nice if you can clearly state the differences.

4- "four encoder-layers in the encoder, and three decoder-layers in the decoder.": how did you choose the numbers of layers here? Same question for all the hyperparameters described under "General Setup" in Section 5.3.

5- Section 4.3.1 The definition of $l(x_i, e_{ia})$ after equation (5) is different from how $l(x_i)$ is used (with a single argument) in (5). What is the definition of $e_{ia}$? Also, can you clarify what you mean by "is applied to specify the polarity of each variable"?

6- Table 3: the time difference with PDP is clearly implementation-dependent. Can you comment on the significance of these results? Also, it is impossible to guess how many learnable parameters PDP and your model have; can you add that information to the table?

Minor:
- "The modern approach is trending to solve CSP through neural symbolic methods." --> "One modern, trendy approach to solving CSP is through neural symbolic methods."
- Unclear what this sentence means: "Hence, the resulted model is bounded by the greedy strategy, which is generally sub-optimal."
- "The adjacent matrix" --> "The adjacency matrix"
- "literals forming a partition different from" --> "literals forming a partition are different from"
- "a highly paralleled message-passing" --> "a highly parallelized message-passing"
- Page 2 para 1, missing full stop in the last line.
- Figure 2(b), the annotations "Encoder" and "Decoder" in the light blue boxes seem to be interchanged

---

### Official Review · AnonReviewer4 · 2020-11-01
**Promosing idea but overly general description and missing key test results**

**Rating:** 4
**Confidence:** 4

**Review:**

This paper presents Heterogeneous Graph Transformer (HGT), a new architecture that combines useful properties of GNNs and Transformers to design HGT for combinatorial reasoning problems, particularly Boolean Satisfiability (SAT). The idea of combining these two powerful models is appealing. The motivation behind considering both homogenous attention (between two literals, or between two clauses) and heterogenous attention (between a literal and a clause) seems new and interesting too. However, there are several missing pieces in the paper in the present form, as outlined below.

The closest existing work, in terms of model architecture, appears to be Graph Attention Network or GATConv (2017). However, the first time this work is mentioned is quite late -- in section 4.2. I'm not sure why. I would have expected to see this relationship mentioned in the intro or related work sections. Without it, the proposal seems more novel than it actually is.

I also find the title, Transformers Satisfy, too distant from the proposed technique, which is a combination of heterogeneous GNNs and attention mechanism, extending GATconv. Why the emphasis on Transformers? They are a particular way of using an all-to-all attention mechanism, which isn't quite what the proposed approach uses.

Similarly, the emphasis of the abstract and intro on solving CSPs too broad for what's supported in the paper. After the sentence "A CSP can be formulated as a CNF formula" on page 2, the rest of the formulation as well as all experiments are for the SAT problem. It would be better to frame the work as a new approach for *satisfiable* instances of SAT. It's ok to mention that it can be generalized to CSPs, but the focus is best left on SAT.  (I should note that even within SAT, the focus here is on 4 families of random instances, leaving out a whole universe of hand-crafted as well as application / "industrial" instances used to evaluate SAT solvers.)

The claim in the 2nd paragraph of the intro, that conventional CSP solvers rely on handcrafted heuristics and hence "the resuling model is bounded by the greedy strategy, which is generally sub-optimal" is misleading. It suggests that existing and proposed neural methods are closer to being optimal than conventional CSP solvers, which cannot be farther from the truth! The fact is, neural methods for CSP and SAT are one to two decades behind conventional solvers in terms of the scale of problems they can convincingly solve. It's an important research area, but not one that has yet even come close to conventional solvers.

The presentation is difficult to follow in places, as it leaves many important details unclear. E.g., section 3.3 tries to give an intuitive connection to Transformers, but relies on specific prior knowledge of queries, keys, similarity computation, etc., in transformers or attention mechanisms, which is left undescribed. I would suggest cutting down some standard details about CSPs or factor graphs, to instead focus on newer concepts.

In secion 4.3.1, the choice of v_i = floor(x_i / (0.5 + eps)) appears unusual and unnecessarily reliant on the parameter epsilon. Why not simply use v_i = floor(x_i + 0.5), the typical way to implement rounding?

Later in that same section, I did not follow the intuition behind the regularizer term in Eqn (8). The motivation was to shift the value of each x_i away from 0.5 and towards either 0 or 1. Presumably it doesn't matter whether x_i goes towards 0 or towards 1, as long as the regularizer pushes it either way. How does the ReLU term achieve this?  (for one, it is asymmetric while the motivation was a symmetric move away from 0.5)

In the experiments, it's not clear what exactly are competing approaches trained on. Is it on the same distribution as the proposed model, or something else? Were any hyperparameters of baseline approaches tuned during training?

I'm not sure what's the intended empirical support for the claim just before the start of 5.2, "Models traind on these graphs generalize to other CSP problems, dataset, and larger graphs." Is there a particular experiment you are referring to?

Importaintly, the experiment section is completely missing accuracy results on the Test set when comparing to RLSAT and PDP. Sure, training accuracy given some intuition, but it's impossible to judge the value of the method without comparing accuracy on unseen Test instances.

---

### Decision · Program_Chairs · 2021-01-07
**Final Decision**

**Decision:**

Reject

**Comment:**

This paper presents a new graph neural network (GNN) architecture with attention and with applications to Boolean satisfiability.

The reviewers expressed concerns over various aspects of the paper such as a need for better ablations and an analysis of the difficulty level of the SAT problems used in evaluation.  No rebuttal was provided.